

# An improved image processing algorithm for visual characteristics in graphic design

Huiying Zhou

School of Arts and Design, Mudanjiang Normal University, Mudanjiang, China

## ABSTRACT

Drawing the clothing plan is an essential part of the clothing industry. However, the irregular shape of clothing, strong deformability and sensitivity to light make the fast and accurate realization of clothing image retrieval a very challenging problem. The successful application of the Transformer in image recognition shows the application potential of the Transformer in the image field. This article proposes an efficient and improved clothing plan based on ResNet-50. Firstly, in the feature extraction section, the ResNet-50 network structure embedded in the Transformer module is used to improve the network's receptive field range and feature extraction ability. Secondly, dense jump connections are added to the ResNet-50 upsampling process, making full use of feature extraction information at each stage, further improving the quality of the generated image. The network consists of three steps: the sketch stage, which aims to predict the color distribution of clothing and obtain watercolor images without gradients and shadows. The second is the thinning stage, which refines the watercolor image into a clothing image with light and shadow effect; The third is the optimization stage, which combines the outputs of the first two stages to optimize the generation quality further. The experimental results show that the improved network's IS and first input delay (FID) scores are 4.592 and 1.506, respectively. High-quality clothing images can be generated only by inputting line drawings and a few color points. Compared with the existing methods, the image generated by this network has excellent advantages in realism and accuracy. This method can combine various feature information of images, improve retrieval accuracy, has strong robustness and practicability, and can provide a reference for the daily work of fashion designers.

## INTRODUCTION

Drawing a clothing plan is an essential part of the clothing industry. Clothing renderings focus on the specific form and design details to accurately grasp the designer's design intention in garment production (*Xu et al., 2023*). The first step of drawing the effect diagram is to sketch clothing, and the second step is to paint, that is, to describe the surface texture and texture of clothing cloth and reflect the dressing effect of clothing (*Fang et al., 2023*). The process of coloring is complicated and requires designers' creativity and painting skills. With the development and maturity of computer technology, researchers began to mine clothing-related information from clothing plane image data, in which line drawings and color points are important clothing information, which is composed of

Corresponding author
Huiying Zhou, jwclsl@163.com

clothing structure, color, texture, and other elements, and can represent the overall visual experience of clothing (*Zhou et al., 2022a*). Therefore, how to use image processing technology to extract various elements from clothing images to help clothing designers grasp the trend and draw a more stylish clothing plan has gradually become a research hotspot (*Xu et al., 2022*).

The mainstream clothing design and painting software includes PhotoShop, Illustrator, CorelDraw, Freehand, *etc.* (*Saranya & Geetha, 2022*). Compared with traditional hand-drawn methods, this software provides various image editing tools, which improve the drawing efficiency and reduce the trial and error cost of coloring. Despite the software's help, the clothing plan's drawing process is still very complicated, and it needs to rely on manual design, which requires high human resources and time costs (*Zhou et al., 2022b*).

In recent years, the convolutional neural network (CNN) has been widely used in image processing, which will cascade representation and classification learning. It can adaptively mine useful feature information from images through a backpropagation algorithm (*Yu et al., 2021*). There are two kinds of CNN-based clothing image processing methods. One is the clothing component-based method (*Mir, Alldieck & Ponsmoll, 2020*) represented by CD-CNNs (component dependent convective neural networks). It uses a support vector machine to process and classify the extracted features. Its deployment of a feature extraction model based on clothing components requires various supervision information related to clothing components. The whole model has high complexity, many parameters, and much calculation. The second is the end-to-end method based on the global image. This method takes the entire image as the input of CNN and automatically learns the characteristics of clothing style (*Gao & Han, 2020*). For example, *Aoki et al. (2019)* used PSPNet to locate the clothing area in the image scene and recognized the clothing style through ResNet50. *Takagi et al. (2017)* collected 13,126 clothing images covering 14 styles, such as dresses, classics, and Lolita, and tested the performance of clothing style recognition and generation using five general image recognition networks such as VGG (visual geometry group). *Liu et al. (2016)* put forward the DeepFashion data set, which improved the traditional deep neural network and enabled it to perform many tasks such as clothing classification, key point location, and retrieval. *Ge et al. (2019)* constructed the DeepFashion data set and proposed the Match R-CNN network for clothing categories, key point detection, segmentation, and generation. *Wang et al. (2018)* and *Yu et al. (2021)* used dynamic dependence and symmetry to encode clothing images and introduced a bidirectional CNN network to model clothing images for feature extraction and key point detection. *Lin et al. (2016)* extracted the pixels of the input image based on the depth confidence network and realized the classification and generation of clothing images. *Tan et al. (2020)* fused the ELU and ReLU functions with the Xception network and applied them to the clothing image processing, achieving a good image generation effect. *Chen & Han (2018)* proposed ResNet and VGG networks based on transfer learning, which realized the processing and classification of clothing images. *Liu, Luo & Dong (2019)* proposed a hierarchical classification model based on ResNet and verified the processing performance of the model on the Fashion Mnist data set. *Di (2020)* proposed a clothing image processing model based on MobileNetV2, The excellent performance of the

MobileNetV2 network is proved on the Fashion Mnist clothing data set. *Cychnerski et al. (2017)* proposed a clothing categories detection system that can judge five attribute tags on DeepFashion data sets. The clothing mentioned above image processing methods based on CNN has completed the tasks of classification, generation, and retrieval of clothing images, which can help clothing designers to reduce the burden of drawing clothing plans to some extent.

At present, CNN-based methods have achieved excellent performance in clothing. However, these methods can not extract fine-grained features in clothing images. Also, it is challenging to learn global semantic information and semantic information with relatively far pixels, which leads to poor feature extraction effect of clothing images by the network. Recently, inspired by the great success of Transformers in natural language processing, researchers plan to introduce Transformer into the field of image processing (*Dosovitskiy et al., 2020*). For example, *Dosovitskiy et al. (2020)* proposed to use Vision Transformer for image recognition and used a two-dimensional image block sequence with location information as input to pre-train on a large data set, with performance equivalent to that based on CNN. *Touvron et al. (2021)* put forward the data-efficient image Transformer, which enables the Transformer to be trained on medium-sized data sets. A more robust transformer can be obtained by combining the knowledge distillation method. *Park & Lee (2022)* designed a layered Swin Transformer, which was used as the backbone network for image feature extraction, and made significant progress in image processing. It can be seen that the successful application of the Transformer in image recognition shows the application potential of the Transformer in the image field. Therefore, we can consider combining CNN and Transformer to extract clothing image features to achieve better processing performance when processing clothing images.

To achieve more accurate and efficient feature extraction of clothing images and help clothing designers draw more stylish clothing plans, this article proposes a clothing image generation network based on Transformer and CNN, which can automatically generate realistic clothing renderings only by specifying a few color points on clothing line drawings. The specific innovations are as follows:

(1) Effectively integrate ResNet-50 and Transformer, obtain local information of clothing images through ResNet-50, and Transformer obtains global information, further improving the network's ability to extract different features of clothing images.

(2) Introducing dense skip connections in ResNet-50 further improves the ability of the network to utilize feature information at various levels, completing more accurate clothing image generation tasks.

## CLOTHING IMAGE GENERATION NETWORK

This article proposes a clothing image generation network based on ResNet-50 and Transformer. Users only need to input a clothing line drawing and specify a few color points on the line drawing to generate realistic clothing images according to their design intentions.

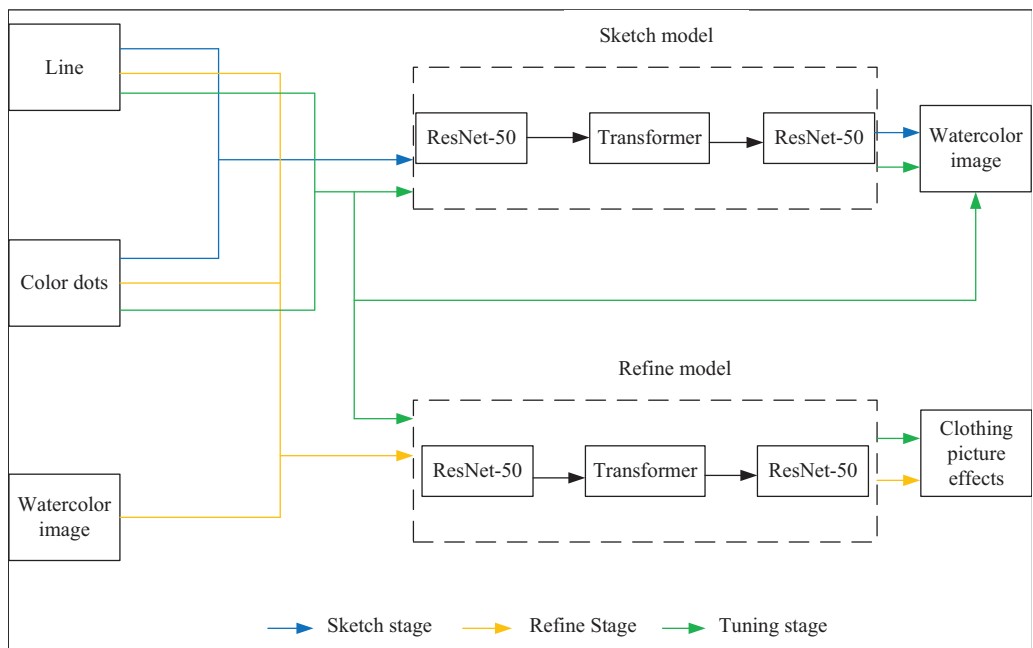

**Figure 1 Schematic diagram of the clothing image generation network.**

Two deep learning network models based on ResNet-50-Transformer hybrid architecture are used: the sketch and refinement models. The training process is divided into three stages: sketch, refinement, and optimization, as shown in Fig. 1.

(1) Sketch stage: the input data in this stage are line drawings and color points, and the target is watercolor clothing images. Watercolor image is characterized by color block distribution, without gradient and light and shadow effects. After training, the sketch model can generate watercolor clothing images to obtain approximate color distribution information.

(2) Refinement stage: The input data in this stage are watercolor images, line drawings and color points, and the generation target is clothing images. After obtaining the approximate color distribution, the refined model can further optimize the generation effect and generate realistic clothing images after this training stage. The input data of line drawings and color points help refine the model and correct the unreasonable generation results in the sketch stage.

(3) Optimization stage: Both sketch and refinement models are trained simultaneously in this stage. The watercolor image generated by the sketch model, line drawings, and color points are taken as the thinning model's input, and the thinning model and the training loss term of the thinning model are taken as a part of the loss term of the sketch model. This stage aims to fine-tune the parameters of the converged sketch model and refined model so that the input and output data of the two models can be more coordinated and the generation quality can be further optimized.

After the model is trained, it is only necessary to input the line draft and color points into the sketch model and then input the output of the sketch model, line draft, and color points into the refined model to generate the clothing effect map.

## Network structure

The sketch and refinement models have different input dimensions, but the other structures are identical. They comprise a ResNet-50, a Transformer, and a Cascaded Upsampler (CUP).

### ResNet-50

Firstly, the local features of the input image are extracted by down-sampling using ResNet-50, which is divided into three down-sampling stages. See Table 1 for the parameters of the down-sampling module. Each module consists of multiple Bottleneck structures repeatedly superimposed. In the Bottleneck part, the number of channels is firstly reduced by a $1 \times 1$ convolution, and then the ordinary $3 \times 3$ convolution in the middle is convoluted. The structure of the Bottleneck is shown in Fig. 2.

In Fig. 2, $d$ represents the number of channels of the feature map.

### Transformer

The input feature map is analyzed $x$ marks the position and divides it into an image block. $\left\{ x_p^i \in R^{p^2 \cdot C} | i = 1, \ldots, N \right\}$, where the size of each image block is $P \times P$, the number of image blocks is $N = \dfrac{HW}{P^2}$, the number of image blocks is also the length of the input image block sequence.

Trainable linear mapping is used to map vectorized image blocks. $xp$ to a $D$ dimension coding space. To encode the spatial information of an image block, specific position information is added to the image block sequence to retain its position information, as shown in Formula (1).

$$z0 = \left[ x_p^1 E; x_p^2 E; \ldots; x_p^N E \right] + Epos \tag{1}$$

where $E \in R^{(p^2 \cdot C) \times D}$ represents the coding mapping of the image block, $Epos \in R^{N \times D}$ represents the location information code.

The transformer coding structure consists of $L$ multi-self-attention (MSA), and multi-layer perception (MLP). Among them, MSA is a self-attention module with multiple branches, each branch represents a head, and the hidden layer and output layer of multi-layer perceptron are all neural networks with full connection layer. The output of the $\eta$-th layer in the coding structure is represented by Eqs. (2) and (3).

$$z'\eta = MSA(LN(z\eta - 1)) + z\eta - 1 \tag{2}$$
$$z\eta = MLP(LN(z'\eta)) + z'\eta \tag{3}$$

where $LN(.)$ represents layer standardization operation, $z\eta$ is an encoded image representation, and the structure of the Transformer module is shown in Fig. 3.

**Table 1 ResNet-50 structure.**

| Convolution part | Output size | 50 floors |
|---|---|---|
| Convolution layer | 128 × 128 | 7 × 7, 64, stride2 |
| Pool layer | 64 × 64 | 3 × 3, max pool, stride2 |
| The first stage | 64 × 64 | $\begin{bmatrix} 1\times1,64 \\ 3\times3,64 \\ 1\times1,256 \end{bmatrix}\times3$ |
| The second stage | 32 × 32 | $\begin{bmatrix} 1\times1,128 \\ 3\times3,128 \\ 1\times1,512 \end{bmatrix}\times3$ |
| The third stage | 16 × 16 | $\begin{bmatrix} 1\times1,256 \\ 3\times3,256 \\ 1\times1,1024 \end{bmatrix}\times3$ |

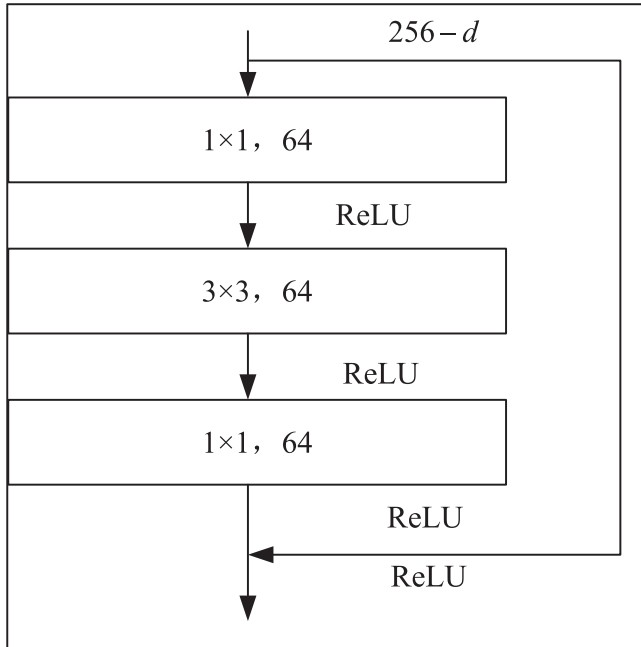

**Figure 2 Bottleneck structure.**     

## Decoder

This article introduces CUP into the decoding part, composed of multiple up-sampling and multi-layer jump connections to decode the hidden features to get the final generated image (*Yu et al., 2021*). After the sequence of hidden features $ZL \in R^{\frac{HW}{P^2} \times D}$ is reconstructed $\frac{H}{P} \times \frac{W}{P} \times D$, the up-sampling is realized by cascading multiple up-sampling

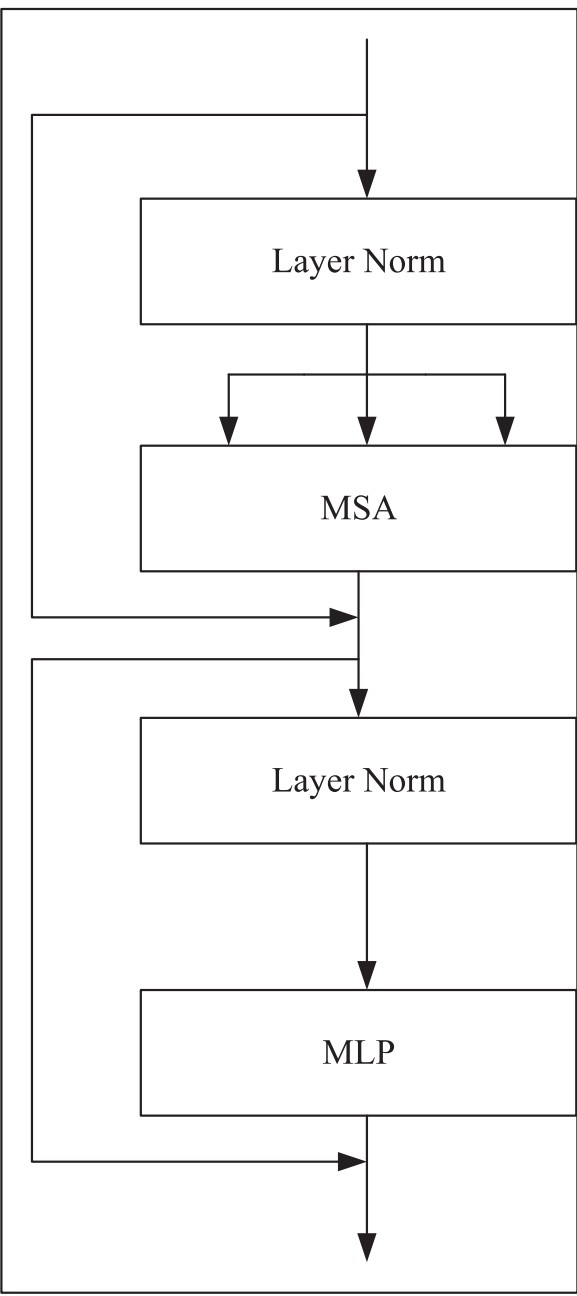

**Figure 3 Transformer coding structure.**

blocks and skip connections. Each up-sampling block is successively composed of feature map concatenation, convolution function and ReLU activation function to achieve complete resolution.

The skip connection in the CUP is used to fuse the multi-scale features of the encoder with the upsampling features. However, in the first stage, the down-sampling only extracts the shallow features of the image, and it will lose a lot of information if it is directly fused with the last feature of the up-sampling. Hence, this article adds a dense jump connection to connect every scale in the horizontal direction. After the two feature maps are spliced by

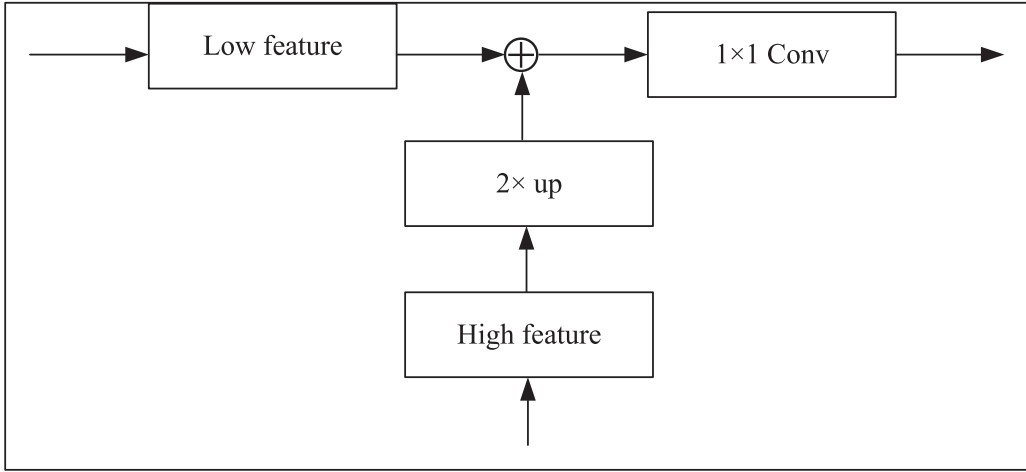

**Figure 4 Feature splicing structure.**

the bilinear interpolation upsampling method, the number of channels is halved by $1 \times 1$ convolution. The specific operation is shown in Fig. 4.

## Discriminator

The discriminator is essential for the clothing image generation network. The discriminator used in this article consists of a simple multi-layer convolution and a fully connected layer at the tail. It is a discriminator network based on reference (*Zhang et al., 2019*). All its batch normalization layers are removed in this article. This modification is because part of the loss function used in this article is based on the reference (*Gulrajani et al., 2017*), and batch normalization in the discriminator may destroy this part of the loss function. *Lipschitz* continuity.

## Loss function

The total loss items used in the training process are:

$$\ell D = \lambda fea\ell fea + \lambda rec\ell rec + \lambda adv\ell adv \tag{4}$$

$$\ell R = \lambda fea\ell fea + \lambda rec\ell rec + \lambda adv\ell adv \tag{5}$$

$$\ell T = \lambda R\ell R + \lambda fea\ell fea + \lambda rec\ell rec \tag{6}$$

where $\ell D$ and $\ell R$ are loss items used in the sketch stage and refinement stage, respectively; $\ell T$ and $\ell R$ represent the optimization stage, which is used to limit the loss items of the sketch model and refinement model, respectively; $\ell rec$, $\ell fea$ and $\ell adv$ are reconstruction loss, feature loss, and confrontation loss. The specific calculation method for these losses is as follows:

(1) Reconstruction loss $\ell rec$ is the loss between the real and generated images *Igen*.

$$\ell rec = \frac{1}{CHW} \|Igt - Igen\|1 \tag{7}$$

**Table 2 Model training parameters.**

| Parameter name | Parameter value |
| --- | --- |
| Optimizer | AdamW |
| Learning rate | 1e−4 |
| Epoch | 2,000 |
| CPU | E3-1230v2 |
| GPU | GTX1660s |
| Memory | 16G |
| OS | Windows10 |

where *CHW* is the size of the image; *C*, *H*, *W* are the number of channels, height and width of the image.

(2) Feature loss $\ell fea$ has been proven effective in improving visual quality and has been applied to some image-generation methods (*Liu, Luo & Dong, 2019*). A pre-trained large-scale classification network often can image semantic information perception, so narrowing the distance between the predicted image and the target image in the feature space can improve the authenticity and accuracy of image generation. The characteristic loss used in this article is:

$$\ell fea = \frac{1}{CiHiWi} \left\| \phi_i^{vgg}(Igt) - \phi_i^{vgg}(Igen) \right\|_2^2 \tag{9}$$

where $\phi_i^{vgg}$ represents the *i*-th characteristic diagram of the ResNet-50 layer; *CiHiWi* represents the size of the feature map of the *i*-th layer.

(3) GAN losses $\ell adv$ restrict the discriminator in the network. The loss term of the discriminator is calculated by the difference between the false image generated by the generator and the real image, and the discriminator gives the loss term of the generator. In order to train more stable and better performance, we use gradient normalization punishment instead of weight clipping here. The loss item can be written as:

$$\ell Dis = \underset{\tilde{x} \sim Pg}{E}[D(\tilde{x})] - \underset{\tilde{x} \sim Pr}{E}[D(x)] + \lambda \underset{\tilde{x} \sim P\hat{x}}{E}\left[(\|\nabla \hat{x} D(\hat{x})\|_2 - 1)^2\right] \tag{10}$$

$$\ell adv = \ell Gen = - \underset{\tilde{x} \sim Pg}{E}[D(\tilde{x})] \tag{11}$$

where $\ell_{Dis}$ is the loss term of the discriminator, and $\ell_{Gen}$ is the loss item of the generator.

## EXPERIMENT AND ANALYSIS

### Data set and parameter settings

In this article, 5,000 clothing renderings were collected from the official website of fast fashion consumer brands such as Uniqlo and Muji, and the OpenCV image processing library standardized the images. See Table 2 for the training process parameter settings. After the model training is completed, the line draft of the clothing image is obtained by

PhotoShop batch processing. Then the line draft data set can be generated by inputting it into the network. All subsequent experiments were conducted on a unified parameter setting and self built dataset.

## Comparison of different methods

Three quantitative evaluation indexes are used to quantitatively evaluate the authenticity and accuracy of clothing images generated by PSPNet (*Aoki et al., 2019*), VGG (*Takagi et al., 2017*), ResNet (*Takagi et al., 2017*), UNet (*Zhang et al., 2022*), Attention-UNet (*Zhu, Shu & Zhang, 2022*), GAN (*Chen & Han, 2018*), Transformer (*Liu, Luo & Dong, 2019*) and this method.

(1) Authenticity evaluation: The experiment adopts human perceptual research, (HPR). Given ten groups of images generated by each method, 20 volunteers must choose the most accurate method from each group based on their first impressions. If a volunteer selects each method, one point will be scored, and the final score will be averaged.

(2) Accuracy evaluation: Inception score (IS) and Frechet Inception Distance score (FID) (*Cychnerski et al., 2017*), which are widely used in the generation model, are adopted in the experiment. Lower FID and higher IS represent more accurate realistic images with higher quality. The specific formula of IS is as follows.

$$IS(G) = \exp(Ex \sim pgDKL(p(y|x)\|p(y)))\qquad(12)$$

where, $Ex \sim pg$ represents traversing all generated samples and averaging them. $DKL$ represents the approximation between different generated samples. $p(y|x)$ represents the probability distribution that the picture $x$ belongs to all categories. $p(y)$ represents an edge probability of $p(y|x)$.

The specific formula of FID is as follows.

$$FID = \|\mu r - \mu g\|^2 + Tr\left(\left(\sum r + \sum g\right) - 2\left(\sum r \sum g\right)^{\frac{1}{2}}\right)\qquad(13)$$

where, $\mu r$ represents the feature mean value of a real image. $\mu g$ represents the feature mean value of the generated image. $\sum r$ represents the covariance matrix of a real image. $\sum g$ represents a covariance matrix for generating images. $Tr$ represents the generation trajectory of the image.

Table 3 lists the quantitative scores of each method. Our method has achieved the best results in three evaluation indexes compared with other methods.

## Ablation experiment

This research also compares the ablation experiment with the pure ResNet-50 network. To observe and analyze the information processing process of each module, the network in this article is split and reorganized. The data from relevant comparative experiments are shown in Table 4.

Compared with the ResNet-50 network, our model combines CUP and Transformer. The accuracy of the proposed model is much better. During the experiment, although

**Table 3 Quantitative comparison between this method and other methods.**

| Methods | HPR | IS | FID |
|---|---|---|---|
| PSPNet | 0.0 | 3.852 | 5.790 |
| VGG | 1.3 | 4.266 | 2.340 |
| ResNet | 1.0 | 4.133 | 2.464 |
| Unet | 1.7 | 4.287 | 2.191 |
| Attention-UNet | 0.8 | 4.174 | 2.419 |
| GAN | 1.9 | 4.304 | 2.085 |
| Transformer | 4.4 | 4.427 | 1.872 |
| Ours | 5.5 | 4.592 | 1.506 |

**Table 4 Functions of transformer and CUP module.**

| Methods | Batch Size | Training time | Memory occupancy | IS | FID |
|---|---|---|---|---|---|
| With CUP and transformer | 4 | 365.2 s | 2.223G | 4.592 | 1.506 |
| Without CUP and transformer | 4 | 1,204.6 s | 7.854G | 4.989 | 1.367 |

**Table 5 Ablation experiment of transformer layer number.**

| Number of plies | Batch size | IS | FID |
|---|---|---|---|
| 2 | 4 | 4.011 | 1.982 |
| 8 | 4 | 4.236 | 1.774 |
| 12 | 4 | 4.592 | 1.506 |

Transformer and CUP modules compressed the feature map, the accuracy of the image generation IS decreased by 8.138%, and the FID increased by 9.437%. Still, the video memory occupation of 71.696% and the training time of 69.683% are reduced. This shows that Transformer and CUP modules can effectively reduce the calculation, play a role in optimizing the model or improve the quality of image generation.

To further verify the effectiveness of the network, the ablation experiments of different numbers of Transformer layers are continued, and the effects of three other numbers of Transformer layers are carried out, which are two, eight and 12 layers, respectively. Table 5 shows the detailed results. Other parameters remain unchanged during the experiment except for the number of Transformer layers.

The experimental results show that with the increase of Transformer layers, the IS score of the network increases and the FID score decreases, which further verifies the effectiveness of the Transformer structure.

## Robustness experiment

In practical application, due to the limitation of shooting conditions, there are disturbances such as rotation, low resolution, and illumination in the imaged clothing image. Therefore,

**Table 6  Results of robustness experiment.**

| | Parameter range | Top-1 accuracy/% | |
|---|---|---|---|
| | | Resnet-50 | Ours |
| Rotation angle | (−5 degrees, 5 degrees) | 74.1 | 75.2 |
| | (−30 degrees, 30 degrees) | 72.7 | 73.4 |
| | (−45 degrees, 45 degrees) | 71.1 | 72.6 |
| Image size | (202,204) | 72.6 | 73.4 |
| | (179,224) | 71.3 | 72.5 |
| | (157,224) | 70.6 | 71.0 |
| Brightness factor | (0.9,1.1) | 74.0 | 75.4 |
| | (0.8,1.2) | 73.3 | 74.6 |
| | (0.7,1.3) | 73.1 | 74.1 |

the robustness of the clothing plane image generation network is one of the requirements for practical application. The robustness of the proposed network and the traditional ResNet-50 are tested. Ten images were randomly selected from the data set for rotation transformation, scaling transformation, and brightness transformation, respectively, and the random range of each transformation parameter was increased in turn. The 10 images were tested five times, and the generation accuracy was taken as the average of five tests. The results of the robustness test are shown in Table 6. Among them, the rotation angle refers to the included angle between the rotated image and the original image, the image size refers to the spatial resolution of the image after scale transformation, and the brightness factor refers to the overall brightness gain of the image after brightness transformation compared with the original image (*Saranya & Geetha, 2022*).

From Table 6, the generation accuracy of the network in this article decreases slightly, which is better than the traditional ResNet-50, indicating that the method in this article has good robustness against three kinds of disturbances. The accuracy will drop sharply when the range of random parameters is further expanded. At this time, it is necessary to improve the robustness of the network by data expansion or disturbance parameter estimation.

## Discussion

Suppose a fashion designer can get one or more corresponding real images of clothing by drawing a rough sketch of apparel. In that case, it will significantly increase the inspiration of fashion designers and greatly reduce their workload. This article uses the advantages of CNN and Transformer to generate clothing images. The model is a generic network that integrates the ideas of Transformer, ResNet, and CUP. Like other network structures, the network consists of the encoder, decoder, and hop connection. The encoder is built based on ResNet-50 and Transformer. After a multi-layer CNN feature extraction structure, the input clothing image is divided into non-overlapping image blocks. The image blocks are combined into image block sequences and input to the encoder based on Transformer to learn global features. Then enter the decoding layer based on CNN to up-sample the

extracted multi-level features, fuse the multi-scale features in the down-sampling process through jump connection, and restore the feature map to approximate the resolution size of the input image to complete the clothing image generation at the pixel level. Through a series of experiments, it can be seen that the work done in this article has a certain degree of innovation. The combination of ResNet-50 with jump connections and Transformer can extract more clothing features, and can generate different clothing real images through simple clothing sketches, which helps improve the design efficiency and design philosophy of clothing designers.

## CONCLUSION

This research proposes a new clothing image generation network, which combines the advantages of CNN and Transformer to obtain information from different dimensions and realizes the task of generating realistic clothing renderings through three stages: sketching, refining, and optimization, which can provide more design inspiration for clothing designers and complete diversified clothing plan drawing. In the experimental comparison, this method surpasses similar methods in different evaluation indexes and can generate a more realistic and clear clothing plan. Future work will consider further improving the performance and speed of the network, for example, finding a more efficient way to integrate CNN and Transformer. In addition, the data set needs to be expanded to generate better results.

### Funding

This work was supported by the Heilongjiang Provincial Office of Philosophy and Social Sciences "The Practice Research of Public Art in Revitalizing Longjiang Tourism and Creating Characteristic City IP in the We Media Era", project number: 21YSE389. This work was also supported by Key research topics of the 20th National Congress of the Communist Party of China for Education Science Planning in Heilongjiang Province. Project: Research on Integrating the Cultivation of Art Talents into the Mass Entrepreneurship and Innovation Construction System Guided by the Spirit of the 20th National Congress of the Communist Party of China project number: GJE1422132. The funders had no role in study design, data collection and analysis, decision to publish, or preparation of the manuscript.

### Grant Disclosures

The following grant information was disclosed by the authors:
Heilongjiang Provincial Office of Philosophy and Social Sciences "The Practice Research of Public Art in Revitalizing Longjiang Tourism and Creating Characteristic City IP in the We Media Era": 21YSE389.
Key research topics of the 20th National Congress of the Communist Party of China for Education Science Planning in Heilongjiang Province: GJE1422132.

## Competing Interests

The authors declare that they have no competing interests.

## Author Contributions

- Huiying Zhou conceived and designed the experiments, performed the experiments, analyzed the data, performed the computation work, prepared figures and/or tables, authored or reviewed drafts of the article, and approved the final draft.

## Data Availability

The code is available at the Supplemental File.

The data are available at at Kaggle: https://www.kaggle.com/datasets/agrigorev/clothing-dataset-full (CC0 license).

## Supplemental Information

Supplemental information for this article can be found online at http://dx.doi.org/10.7717/peerj-cs.1372#supplemental-information.

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
