# Peer review of "An improved image processing algorithm for visual characteristics in graphic design"

_PeerJ Computer Science, doi:10.7717/peerj-cs.1372_

## Round 0.1 · original submission · Major Revisions

A couple of major changes are required as suggested by the experts, therefore, please carefully revise and resubmit.

·

Basic reporting

This paper proposes a new clothing image generation network, which combines the advantages of CNN and Transformer to obtain different dimensions of information. The generation task of realistic clothing effect drawing can be realized through the three stages of sketch, refinement and optimization, which can provide more design inspiration for clothing designers and complete diversified clothing plan drawing. In the experimental comparison, the proposed method surpasses similar methods in different evaluation indexes, and can generate more realistic and clearer clothing plan. However, there are some questions to be clarified for the exact contributions and the description of the methods achieved in this work as follows.
A. The author should pay attention to elaborate their own innovation achievements,
B. Can not distinguish the achievements of others and their own contribution, so that the innovation point is ambiguous and contradictory
C. The definitions of IS and FID need to be clear. In addition to the abstract, they also appear in the result analysis;
D. In the literature review, the author needs to introduce the methods used in other studies in detail, such as "PSPNet", "R-CNN", etc.;
E. Therefore, the author needs to explain the main innovations of this paper in a more clear and intuitive way;
F. At what stage does Transformer embed take place? sketch stage, refinement stage and optimization stage; Sketch stage, refinement stage and optimization stage;
G. Why only consider the influence of Transformer layer on model training effect, and why not consider the parameter generalization performance of CNN?
H. Other studies should be cited to increase the theoretical background of each of the methods used.
I. Findings should be contextualized in the literature and should be explicit about the added value of the study towards the literature.
J. The English of your manuscript must be improved before resubmission. We strongly suggest that you obtain assistance from a colleague who is well-versed in English or whose native language is English.

Experimental design

This paper proposes a new clothing image generation network, which combines the advantages of CNN and Transformer to obtain different dimensions of information. The generation task of realistic clothing effect drawing can be realized through the three stages of sketch, refinement and optimization, which can provide more design inspiration for clothing designers and complete diversified clothing plan drawing. In the experimental comparison, the proposed method surpasses similar methods in different evaluation indexes, and can generate more realistic and clearer clothing plan. However, there are some questions to be clarified for the exact contributions and the description of the methods achieved in this work as follows.
A. The author should pay attention to elaborate their own innovation achievements,
B. Can not distinguish the achievements of others and their own contribution, so that the innovation point is ambiguous and contradictory
C. The definitions of IS and FID need to be clear. In addition to the abstract, they also appear in the result analysis;
D. In the literature review, the author needs to introduce the methods used in other studies in detail, such as "PSPNet", "R-CNN", etc.;
E. Therefore, the author needs to explain the main innovations of this paper in a more clear and intuitive way;
F. At what stage does Transformer embed take place? sketch stage, refinement stage and optimization stage; Sketch stage, refinement stage and optimization stage;
G. Why only consider the influence of Transformer layer on model training effect, and why not consider the parameter generalization performance of CNN?
H. Other studies should be cited to increase the theoretical background of each of the methods used.
I. Findings should be contextualized in the literature and should be explicit about the added value of the study towards the literature.
J. The English of your manuscript must be improved before resubmission. We strongly suggest that you obtain assistance from a colleague who is well-versed in English or whose native language is English.

Validity of the findings

This paper proposes a new clothing image generation network, which combines the advantages of CNN and Transformer to obtain different dimensions of information. The generation task of realistic clothing effect drawing can be realized through the three stages of sketch, refinement and optimization, which can provide more design inspiration for clothing designers and complete diversified clothing plan drawing. In the experimental comparison, the proposed method surpasses similar methods in different evaluation indexes, and can generate more realistic and clearer clothing plan. However, there are some questions to be clarified for the exact contributions and the description of the methods achieved in this work as follows.
A. The author should pay attention to elaborate their own innovation achievements,
B. Can not distinguish the achievements of others and their own contribution, so that the innovation point is ambiguous and contradictory
C. The definitions of IS and FID need to be clear. In addition to the abstract, they also appear in the result analysis;
D. In the literature review, the author needs to introduce the methods used in other studies in detail, such as "PSPNet", "R-CNN", etc.;
E. Therefore, the author needs to explain the main innovations of this paper in a more clear and intuitive way;
F. At what stage does Transformer embed take place? sketch stage, refinement stage and optimization stage; Sketch stage, refinement stage and optimization stage;
G. Why only consider the influence of Transformer layer on model training effect, and why not consider the parameter generalization performance of CNN?
H. Other studies should be cited to increase the theoretical background of each of the methods used.
I. Findings should be contextualized in the literature and should be explicit about the added value of the study towards the literature.
J. The English of your manuscript must be improved before resubmission. We strongly suggest that you obtain assistance from a colleague who is well-versed in English or whose native language is English.

Additional comments

This paper proposes a new clothing image generation network, which combines the advantages of CNN and Transformer to obtain different dimensions of information. The generation task of realistic clothing effect drawing can be realized through the three stages of sketch, refinement and optimization, which can provide more design inspiration for clothing designers and complete diversified clothing plan drawing. In the experimental comparison, the proposed method surpasses similar methods in different evaluation indexes, and can generate more realistic and clearer clothing plan. However, there are some questions to be clarified for the exact contributions and the description of the methods achieved in this work as follows.
A. The author should pay attention to elaborate their own innovation achievements,
B. Can not distinguish the achievements of others and their own contribution, so that the innovation point is ambiguous and contradictory
C. The definitions of IS and FID need to be clear. In addition to the abstract, they also appear in the result analysis;
D. In the literature review, the author needs to introduce the methods used in other studies in detail, such as "PSPNet", "R-CNN", etc.;
E. Therefore, the author needs to explain the main innovations of this paper in a more clear and intuitive way;
F. At what stage does Transformer embed take place? sketch stage, refinement stage and optimization stage; Sketch stage, refinement stage and optimization stage;
G. Why only consider the influence of Transformer layer on model training effect, and why not consider the parameter generalization performance of CNN?
H. Other studies should be cited to increase the theoretical background of each of the methods used.
I. Findings should be contextualized in the literature and should be explicit about the added value of the study towards the literature.
J. The English of your manuscript must be improved before resubmission. We strongly suggest that you obtain assistance from a colleague who is well-versed in English or whose native language is English.

·

Basic reporting

In order to solve the problems of insufficient intelligence level and high requirement on the drawing level and imagination ability of fashion designers, the author proposes an improved ResNet-50 based clothing plane image generation network. ResNet-50 network structure is used for feature extraction, and Transformer module is used to expand receptive field after feature extraction. In the up-sampling process, intensive jump connection is added to make full use of the feature extraction information of each stage, so as to improve the quality of the generated image.
1. This manuscript may be published in this journal after addressing the following comments
2. It is not necessary to divide the phases of the model, but to pay attention to the optimization and improvement of the model in the Abstract;
3. The combination of Transformer and CNN has been shown in many past studies, for example
Karpov P, Godin G, Tetko I V. Transformer-CNN: Swiss knife for QSAR modeling and interpretation[J]. Journal of Cheminformatics, 2020, 12(1): 1-12.
4. Title of Section 2.1 "Overview of methods" needs to be replaced to be more professional;
5. Discriminator needs more introduction.
6. The Rotation angle, Image size and Brightness factor mentioned in Table 6 require more information on the choice of parameters, including references if necessary;
7. Discussion section needs to be a coherent and cohesive set of arguments that take us beyond this study in particular, and help us see the relevance of what the authors have proposed.
8. Author need to contextualize the findings in the literature, and need to be explicit about the added value of your study towards that literature.

Experimental design

The Rotation angle, Image size and Brightness factor mentioned in Table 6 require more information on the choice of parameters, including references if necessary;

Validity of the findings

Author need to contextualize the findings in the literature, and need to be explicit about the added value of your study towards that literature.

Additional comments

NA

---

## Round 0.2 · accepted · Accept

Thanks for revising the manuscript according to the comments of the experts. Good luck

·

Basic reporting

Authors have done requested changes, they have enhanced the paper quality .. So, I suggest to accept the paper in it current form

Experimental design

OK

Validity of the findings

OK

Additional comments

OK

·

Basic reporting

All suggestions are well addressed. I accept the manuscript for publication

Experimental design

All suggestions are well addressed. I accept the manuscript for publication

Validity of the findings

All suggestions are well addressed. I accept the manuscript for publication

Additional comments

All suggestions are well addressed. I accept the manuscript for publication